



**A comprehensive geospatial database of nearly 100,000 reservoirs in China**
Chunqiao Song[1*], Chenyu Fan[1, 2*], Jingying Zhu[1, 2*], Jida Wang[3], Yongwei Sheng[4], Kai Liu[1],
Tan Chen[1], Pengfei Zhan[1, 2], Shuangxiao Luo[1, 2], Linghong Ke[5]
[1] Key Laboratory of Watershed Geographic Sciences, Nanjing Institute of Geography and
Limnology, Chinese Academy of Sciences, Nanjing 210008, China.
[2] University of Chinese Academy of Sciences, Beijing 100049, China
[3] Department of Geography and Geospatial Sciences, Kansas State University, Manhattan, KS
66506, USA.
[4] Department of Geography, University of California, Los Angeles, CA 90095, USA.
[5] College of Hydrology and Water Resources, Hohai University, Nanjing 210098, China.
*Correspondence to* cqsong@niglas.ac.cn, fanchenyu21@mails.ucas.ac.cn, or
zhujingying18@mails.ucas.ac.cn



**Abstract**
With rapid population growth and socioeconomic development over the last century, a great
number of dams/reservoirs have been constructed globally to meet various needs. China has
strong economical and societal demands for constructing dams and reservoirs. The official
statistics reported more than 98,000 dams/reservoirs in China, including nearly 40% of the
world's large dams. Despite the availability of several global-scale dam/reservoir databases
(e.g., the Global Reservoir and Dam database (GRanD), the GlObal geOreferenced Database
of Dams (GOODD), and the Georeferenced global Dams And Reservoirs (GeoDAR)), these
databases have insufficient coverage of reservoirs in China, especially for small and newly
constructed ones. The lack of reservoir information impedes the estimation of water budgets
and evaluation of dam impacts on hydrologic and nutrient fluxes for China and its downstream
countries. Therefore, we presented the China Reservoir Dataset (CRD), which contains 97,435
reservoir polygons as well as fundamental attribute information (e.g., name and storage capacity)
based on existing dam/reservoir products, national basic geographic datasets, multi-source open
map data, and multi-level governmental yearbooks and databases. The reservoirs in compiled
CRD have a total maximum water inundation area of 50,085.21 $km^2$ and a total storage capacity
of about 979.62 Gt (924.96-1060.59 Gt). The quantity of reservoirs decreases from the
southeast to the northwest, and the density hotspots mainly occur in hilly regions and large
plains, with the Yangtze River Basin dominating in reservoir count, area, and storage capacity.
We found that these spatial accumulations of reservoirs are closely related to China's
socioeconomic development and the implementation of major policies. Finally, we discussed
the improvements of CRD in comparison with GOODD, GeoDAR, and GRanD datasets. CRD
has significantly improved the reservoir count, area, and storage capacity in China, especially
for reservoirs smaller than 1 $km^2$. The CRD database provides more comprehensive reservoir
spatial and attribute information and is expected to benefit water resources managements and
the understanding of ecological and environmental impacts of dams across China and its
affected transboundary basins.

## 1 Introduction

Reservoirs and their dams play a crucial role in green energy generation and water resources management. Since the mid-20th century, the ever-growing human demands for water use and hydropower has driven an unprecedented boom in reservoir construction worldwide (Chao et al., 2008; Wada et al., 2017). The dam construction and reservoir impoundment can lead to many potential environmental and socio-economic impacts (Jiang et al., 2018; Zarfl et al., 2019). These concerned consequences mainly include threat to biodiversity and ecosystems (Winemiller et al., 2016), change in the hydrological regime (Zhang et al., 2019; Vörösmarty et al., 2003), degradation of water quality (Zarfl et al., 2019; Barbarossa et al., 2020), modification of the geochemical cycle (Maavara et al., 2020), alternation of the river morphology (Bednarek, 2001; Nilsson and Berggren, 2000; Winemiller et al., 2016; Grill et al., 2019; Latrubesse et al., 2017; Bond and Cottingham, 2008; Nilsson et al., 2005; Wang et al., 2017a; Wang et al., 2013), disturbance in climate regimes (Pekel et al., 2016; Degu et al., 2011; Wang et al., 2017b; Van Manh et al., 2015), migration of human settlement (Tilt et al., 2009), and changes in the land-use patterns (Stoate et al., 2009; Carpenter et al., 2011).

Despite these controversial effects, artificial reservoirs have been constructed widely across many basins of the world, serving a variety of purposes such as hydropower generation, water supply, irrigation, navigation, flood control, recreation, and navigation (Belletti et al., 2020; Biemans et al., 2011; Döll et al., 2009; Grill et al., 2019; Boulange et al., 2021). In addition, reservoirs assist water managers in converting natural flow conditions into flow conditions that meet human demands, which is especially important in locations where water resources are restricted due to the hydrologic seasonality or the growing influences of climate change and variability (Richter et al., 2006).

The solution to balance the benefits and consequences of reservoirs should not be a simple decision on whether or not to construct them. The significant benefits and the additional effects highlight the importance and necessity for a holistic picture of the reservoir distributions and continuous monitoring of them to understand the impacts better. Information and data regarding reservoirs are rather crucial for scientists, practitioners, and policymakers owing to various purposes, for instance, estimation of water budgets and impacts on hydrologic and nutrient fluxes on regional or global scales (Chao et al., 2008; Bakken et al., 2013; Bakken et al., 2016; Popescu et al., 2020; Postel, 2000), water availability projection or flood/drought risk mitigation (Di Baldassarre et al., 2017; Ehsani et al., 2017; Elmer et al., 2012; Veldkamp et al., 2017; Metin et al., 2018), assessment of hydropower station construction (Bertoni et al., 2019; Gernaat et al., 2017; Xu et al., 2013; Moran et al., 2018; Winemiller et al., 2016), and



investigation of biotic disturbance (Latrubesse et al., 2017; Maavara et al., 2020; Dorber et al.,
2020; Sabo et al., 2017). Considering reservoirs in physical models can significantly improve
the modeling performance (Gutenson et al., 2020). The modeling requires knowing a minimum
set of the reservoir characteristics, including their spatial location, abundance, area, and storage
capacity. Besides, the reservoirs are considered a key source of greenhouse gases (GHGs),
partly offsetting the carbon sink of continents (St. Louis et al., 2000; Aufdenkampe et al., 2011;
Barros et al., 2011; Raymond et al., 2013; Deemer et al., 2016). There is thus an increasing
concern about the true GHGs fluxes from reservoirs. Answering these questions requires a
comprehensive database depicting reservoir distributions and properties, especially for
hydropower-boom regions in Asia, South America, and Africa.
China has a strong economical and societal demand for hydroelectric development, flood
control, and agricultural irrigation. In 2007, China's Medium-and Long-Term Plan for
Renewable Energy Development projected constructing 300 GW of gross installed hydropower
capacity by 2020, exceeding the doubled capacity in 2007. The installed hydropower capacity
target has been reset to $420\times10^6$ kW by 2020, representing a 70% increase in 2012. In China,
more than 60% of total water consumption is taken by the agricultural water sector, among
which 90% of the quota is shared by the irrigation water use (Jiang et al., 2018). Therefore,
reservoir construction in China has experienced a drastic growth. The number of Chinese
reservoirs increased slowly after the 1980s and soared to the count of 98,000 around 2015
(MWR, 2016). According to the register of the International Commission on Large Dams
(ICOLD and CIGB, 2011), China possesses nearly 40% of the global large dams (storage
capacity greater than 0.1 Gt). However, little is known on the spatial locations and related
georeferenced information of these constructed reservoirs at the national level for China.
There have been multiple efforts to inventory global reservoirs including those of China. The
most recognized and comprehensive database is the World Register of Dams (WRD), hosted
and maintained by ICOLD, which reports 23,841 dams for China. However, as this database is
not georeferenced, its utility is severely limited. The Global Reservoir and Dam database
(GRanD) (Lehner et al., 2011) was an initiative database that can provide global geospatial
details about reservoirs and their attributes. Its latest version, v1.3, contains 7,320
dams/reservoirs, with a cumulative capacity of 6,881 km³, while only 921 Chinese reservoirs
were included. In recent, the GlObal geOreferenced Database of Dams (GOODD) (Mulligan et
al., 2020) and the Georeferenced global Dams And Reservoirs dataset (GeoDAR) (Wang et al.,
2022) were published, containing more than 38,000 and 20,214 reservoirs in a global scale,
respectively. GOODD was manually digitized from high-resolution Google Earth imagery
whereas GeoDAR was georeferenced from ICOLD WRD with a full harmonization with



GRanD. For the Chinese territory, the GOODD and GeoDAR databases contain 9,238 and
4,859 reservoirs, respectively, still significantly below the scales of WRD and MWR. Given the
lacked information, a comprehensive and spatially-explicit database of reservoirs in China is
required.
This study aims to share, as comprehensively as possible, fundamental open-access information
on reservoirs in China. We have compiled the database based on a variety of data sources,
including the national 1:250,000 public basic geographic database, the Almanac of China's
Water Power, three global reservoir inventories (GeoDAR v1.1, GRanD v1.3, GOODD V1.0),
and other published documents and online maps (e.g., Open Street Map (OSM) and Tianditu
Map). A comparison with GeoDAR, the GRanD, and GOODD was conducted to assess the
database. Our inventory contains significantly more reservoirs than the currently available
databases. This database can provide researchers with basic information on reservoir locations,
spatially-explicit inundation areas, water storage, and related details in China, with the goal of
advancing research on water resources, ecological and environmental consequences, global
change impacts, and socioeconomic sector assessments on a national and worldwide scale.

## 124  2 Data description

### 125  2.1  Multi-source data for compiling national reservoir locations

2.1.1 Existing reservoir or dam databases
Before constructing the reservoir database in China, the data of existing dams and reservoirs
are preliminarily compiled as the basis for determining the location of reservoirs. Existing
dam/reservoir databases containing geographical information are one of the key spatial data
sources for reservoirs, including GRanD, GOODD, GeoDAR, and Future Hydropower Dams
(FHReD).
GRanD is a data product of the Global Water System Project and was released firstly in 2011
(Lehner et al., 2011). GOODD (Mulligan et al., 2020) is a comprehensive global dam database
provided by manual inspection and digitization based on multi-source remote sensing satellite
observations and Google Earth images. FHReD database collects spatial locations of reservoirs
that are currently being built or those are planned in the future (Zarfl et al., 2015). GeoDAR is
a global dam and reservoir geographic database based on the multi-source data fusion and
online geocoding of the ICOLD reservoir records (Wang et al., 2022). The FHReD database
provides information on 3,700 planned and under construction reservoirs worldwide, of which
251 reservoirs are located in China and 97 have been dammed by 2020.
In this study, these above-mentioned databases were used to provide location information on



part of China's reservoirs, particularly those of large size. We integrated the spatial information
of existing reservoirs in China and eliminated duplicate information. This way, the CRD retains
the spatial information of each unique Chinese reservoir in these three global databases.
2.1.2 National basic geographic databases
The national 1:250,000 public basic geographic database covers the whole land area of China
and major islands. Overall, the map elements represent the landscape's situation around 2015.
The database, which contains nine element layers such as waterbody (point, line, and surface
layer), is treated with the security technology of spatial location accuracy and attribute content.
Reservoir information is contained in the waterbody layer provided by the basic topographic
map and the layers of natural place names (notes), most of which have name attributes and
spatial positioning information. Although the national surveying authorities provide the basic
terrain data, the spatial coordinates are biased due to the confidential processing of the map.
Therefore, we carried out rigorous data correction and quality control by referring to the high-
resolution Google Earth imagery. Finally, the database provided the spatial information
references of 27,047 reservoirs for the CRD database.
The Tiandi Map is an online-map system developed by the State Bureau of Surveying and
Mapping of China (https://map.tianditu.gov.cn/, only in Chinese), which provides the
geographic information services in two forms: portal and service interface. It integrates public
geographic information resources from national, provincial, and prefecture (county)-level
mapping and geographic information departments, relevant government departments,
enterprises and institutions, social groups, and the public. In addition, users can use the service
interface to call the authoritative, standard, unified online geographic information
comprehensive service of the Tiandi Map. In this study, the Tiandi Map was mainly used in two
aspects: firstly, as a base map for visual interpretation and supplementing the potentially
missing reservoirs. In this process, we initially identified about 60,000 potential reservoirs;
secondly, the map was used to provide the reservoir name attribute. According to the locations
of the reservoir checked by manual inspection based on the Tiandi Map, the name of the
reservoir was queried by calling its reverse geocoding API.
2.1.3 Open-source map data
Open-source maps such as OSM were another key source of obtaining reservoir locations. OSM
is a platform for users, organizations, or countries worldwide to organize and maintain multi-
source geographic information data. Map vector data is available for download under an open
database license. Due to OSM data's open-source and shared characteristics, the collected
multi-source geographic information data can be used as a supplement to other time-limited



databases. They can better reflect the changes in land surface information promptly. OSM
contains data such as water system, road traffic, natural boundary, land use, and construction.
Water system data provides part of reservoir polygon data with names, mainly compiled
manually by OSM users. Finally, the spatial locations of 89 reservoirs were obtained from the
OSM.

**2.2 Data sources for reservoir inundation area mapping**

Water inundation area is an important indicator of the reservoir and a variable for modeling
reservoir storage capacity. Since the reservoir area is dynamically changing, we considered the
maximum water area of the reservoir over the last several decades in this study. Moreover, the
maximum water area of the reservoir can indirectly reflect its water storage capacity. Therefore,
we merged two water occurrence datasets, the Global Surface Water v1.0 (GSW) and Global
Land Analysis and Discovery (GLAD), to obtain long-term historical maximum water areas of
each of the compiled reservoirs.
GSW is a remote sensing big data computing platform developed by Pekel et al. (2016) using
Google Earth Engine (GEE). Based on all available Landsat 5, 6, 7, and 8 data acquired from
1984 to present, Pekel et al. (2016) used the expert classification system to divide each available
pixel into water bodies and non-water bodies and integrated the results into the data of monthly,
annual, and decadal timescales. The maximum water boundary, water inundation frequency,
water change intensity, water transition, water recurrence, seasonal water, monthly water range,
monthly water recurrence, and annual water range are provided. GLAD is the global water body
map from 1999 to 2019 obtained by Pickens et al. (2020) using GEE remote sensing big data
computing platform based on Landsat 5, 7, and 8 images. The surface water range changes
during this period were highlighted, and the water was classified into several categories based
on water probability, including permanent water area, seasonal water area, lost water area, new
water area, temporary land area, and temporary water area, and high change area.
Considering that both the GSW and GLAD datasets are at 30 m resolution, we also applied
FROM-GLC10 at 10 m resolution based on Sentinel-2 data in 2017 (Gong et al., 2019)  to
handle the incomplete mapping of extremely narrow boundaries for a few reservoirs located in
deep valleys. This database takes the existing land cover data as training samples. It combines
the data of the Space Shuttle Radar Terrain Mission (SRTM) on the GEE big data computing
platform to classify the data by random forest method to obtain the maps of alpine and swamp
areas with overall accuracy loss rate less than 1%. The training samples were classified based
on Landsat 8 original images and eight important indices commonly used in remote sensing
monitoring, such as normalized vegetation index, modified water index, and normalized



building index.

**2.3 Data sources for reservoir storage capacity estimation**

The reservoir storage capacity records were retrieved from various yearbook and documents,
including the Almanac of China's Water Power and other government documents. The Almanac
of China's Water Power is a professional industry yearbook for hydropower in China, providing
detailed information on China's mega reservoirs, including the reservoir location, the dam
purpose, the basin area, the storage capacity, and water level data of various types, and the dam
construction and impoundment time. Other government documents used in the study mainly
include the "List of Persons responsible for the safety of Large reservoirs in China in 2020"
issued by the Ministry of Water Resources, the "List of persons responsible for the safety of
large and medium-sized reservoirs" issued by different provinces and prefectures of China, and
the "List of Reservoirs in Hunan Province" issued by the Water Resources Department of
Hunan Province. The documents provide information on the type and location of the
dam/reservoir and the storage capacity of reservoirs of different sizes. Finally, from the
Almanac of China's Water Power and some other government documents, we collected
authoritative information on the locations and storage capacities of 5,143 reservoirs.

**3 Methodology**

**3.1 Reservoir location extraction**

To build this database, we started with a preliminary compilation of the location information of
Chinese dams and reservoirs from three types of data sources (see Figure 1a). The first type of
sources is the published georeferenced databases for dams and reservoirs, including GRanD,
GOODD, FHReD, and GeoDAR. We combined China's reservoir location information of the
four published dam/reservoir products. After removing duplicates by manual inspection, we
obtained the names and locations of about 7,400 unique reservoirs. The second type of sources
are national basic geographic databases (including the national 1:250,000 public basic
geographic database and Tiandi Map), the Almanac of China's Water Power, and other
government documents. We checked the national 1:250,000 public basic geographic database,
and its drainage layer data and natural place name layer contained most reservoir information.
Here, the Tiandi Map was used a base map for visual interpretation to supplement missing
reservoirs in the national public basic geographic database. Moreover, we made a list of
reservoirs from the Almanac of China's Water Power and documents from local governments,
which only provided the county level address for each reservoir. We then employed the Tiandi
Map geocoding API to query the latitudes and longitudes of these reservoirs. Based on the
second type of data sources, we obtained the location information of about 90,000 reservoirs.



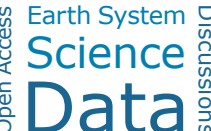

The third type of data sources is open map database, the OSM. From the OSM, we obtained the
location information of 89 reservoirs. After harmonizing the three types of sources, we
concluded with the locations of a total of 97,435 unique reservoirs in China.

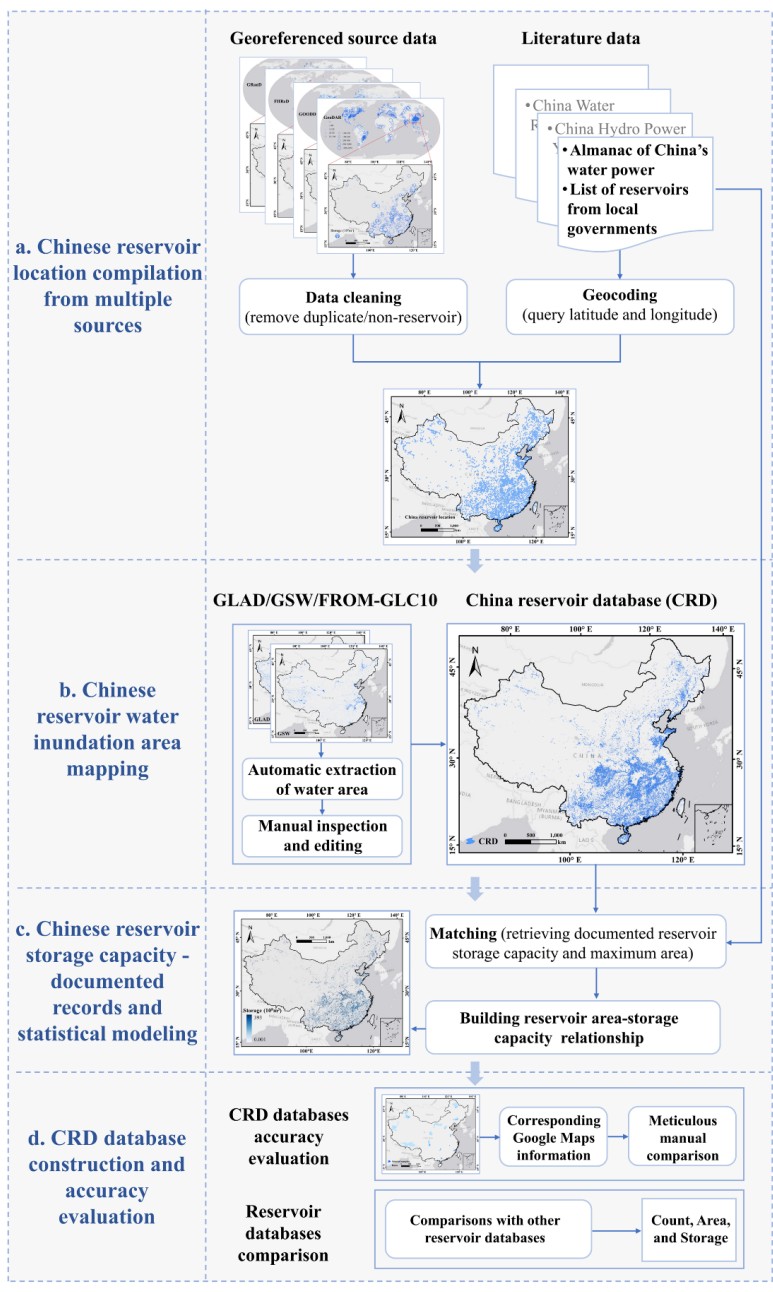


248                    Figure 1. Flow chart of constructing China reservoir database

### 3.2 Reservoir water inundation extent mapping

After determining the spatial location of all reservoirs, we extracted the historical maximum water inundation extent (from the mid-1980s to 2020) of the corresponding reservoirs based on GSW, GLAD, and FROM-GLC10 data (Figure 1b). GSW data can provide the maximum water area of reservoirs with a long-time series from 1984 to 2020. GLAD only maps images over the last 20 years, but it combines Landsat with Sentinel-1 and Sentinel-2 to provide higher temporal resolution to describe ephemeral surface water better. Through comparative inspection, we found that GLAD could describe the water area details more completely for some reservoirs, especially narrow river-channel reservoirs. Therefore, we merged GSW and GLAD datasets to obtain the maximum water area of all reservoirs. In addition, the FROM-GLC10 is based on the Sentinel 10-m resolution imagery data, which can identify relatively small reservoirs (reservoir area smaller than 0.01 $km^2$). Therefore, we also supplemented a few narrow river-channel reservoirs, especially those in mountainous regions of Zhejiang, Fujian, Sichuan, Jiangxi, and Guangxi provinces. The automatically-extracted water masks by intersecting with our compiled reservoir point locations were visually inspected and if necessary, manually edited (such as to separate the reservoir from the river segment) to form quality-controlled reservoir boundaries. Up to now, there are still reservoirs that have not been collected except those identified in Section 3.1. So, all the remaining water bodies were manually checked by overlapping with the Google Earth high-resolution images to minimize the number of missed reservoirs. Finally, a total of 97,435 reservoir polygons were extracted.

For reservoirs without corresponding names, the reverse geocoding API of Tiandi Map was used to query the names of corresponding reservoirs. Here, the reverse geocoding API refers to entering the reservoir's coordinate and then returning the relevant name information of the corresponding reservoir. Eventually, 66,253 reservoirs were identified and supplemented with the name attribute.

### 3.3 Reservoir storage capacity estimation

Reservoir storage capacity is one of the basic information about reservoirs. As shown in Figure 1c, the source of reservoir storage capacity in the CRD database is mainly divided into two types: the recorded values obtained from the yearbook and government documents as mentioned in Section 2.2, and statistical estimations by an empirical model.

According to the yearbook and other documents (Section 2.2), we collected the storage capacity records for 5,143 reservoirs in various sizes, among which 162 Type-I super-large reservoirs (storage capacity greater than 1 Gt), 580 Type-II large reservoirs (0.1-1 Gt), and 4,407 small and medium-sized reservoirs (smaller than 0.1 Gt). As super-large reservoirs (mostly canyon





type reservoirs) tend to have different hypsometric (area-storage relationship) characteristics
from small and medium-sized reservoirs (mostly in plain and hilly areas), we excluded the 742
large reservoirs from model calibration. In addition, we removed 84 reservoirs that do not
conform small and medium-sized reservoirs class (storage capacity smaller than 0.1 Gt). The
statistical relationship between inundation area and storage of a total of 4,323 reservoirs was
established to estimate and supplement the capacity estimation of the remaining unrecorded
small and medium-sized reservoirs. The empirical model was used to fit the storage capacity
and area of the existing recorded reservoirs (Figure 2). The fitting equation is as follows and
the $R^2$ is 0.844.

$$\log_{10} V = 1.096 \times \log_{10} S + 0.349 \tag{1}$$

$$\text{SMAPE} = 100 \times \frac{1}{N} \sum \frac{|\text{observed value - predicted value}|}{(\text{observed value + predicted value})/2} \tag{2}$$

where $V$ represents the reservoir storage capacity in the unit of $m^3$, and $S$ represents the reservoir
maximum area in the unit of $m^2$. We calculated the SMAPE of estimated storage capacity was
biased of 32.62-32.64% at the 95% confidence interval based on the fitted model. Finally, the
recorded values from yearbook and other documents are regarded as the storage capacity of
5,143 reservoirs, totaling about 803.29 Gt. The other 92,292 reservoirs storage capacity were
estimated by using their maximum inundation areas as in equation (1), with a total of 176.33
Gt, ranging from 121.67 Gt to 257.30 Gt. Therefore, the total storage capacity of Chinese
reservoirs is 979.62 Gt (924.96-1060.59 Gt).

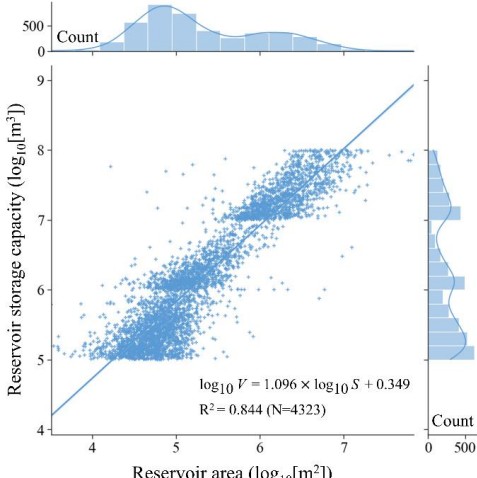


Figure 2. Fitting relationship of area and storage capacity of small and medium-sized reservoirs.
The bars and broken lines in the subgraph respectively represent the count of scattered points

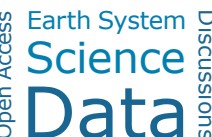

and kernel density in the corresponding interval. The upper and right subplots correspond to the
count of reservoir area and storage capacity values, respectively.
**4 Results**
**4.1 Description of the CRD database**
This database catalogs the location information of 97,435 reservoirs in China, with an
aggregated area of 50,085.21 km$^2$ and an estimated total storage capacity of 979.62 Gt (924.96-
1060.59 Gt). The 5,143 reservoirs in the CRD database were directly derived from the yearbook
and other documents data, accounting for 59% and 82% of the total reservoir area and storage
capacity of the CRD database, respectively. This reservoir information was mainly obtained
through manual compilation. The attributes of the recorded reservoirs include the longitude and
latitude of the reservoir, name, province, prefecture, and county where the reservoir is located,
water area, normal water level, storage capacity, reservoir class, main use, and regulation type
(Table 1). The attributes of all the CRD reservoirs (in all cases) include location information
(longitude, latitude, province, prefecture, and county), inundation area, and estimated storage
capacity, as shown in Table 2.
Table 1. Attributes in the recorded (5,143) reservoirs from yearbook and document data.

| Attribute | Description |
|---|---|
| ID | Reservoir ID in this database (type: integer). |
| Name | Name of the reservoir. |
| Lat | Latitude of the reservoir point (type: float, datum: World Geodetic System (WGS) 1984, unit: °). |
| Lon | Longitude of the reservoir point (type: float, datum: WGS 1984, unit: °). |
| Province | Province in which the reservoir is located. |
| Prefecture | Prefecture in which the reservoir is located. |
| County | County in which the reservoir is located. |
| Area | Maximum water area of the reservoir (unit: km$^2$). |
| Normal elevation | Normal elevation of the reservoir (unit: m). |
| STOR_Recor | Total storage capacity of values from yearbook and literature records (unit: Gt). |
| ResvClass | Reservoir class (1: large Type-I, 2: large Type-II, 3: medium, 4: small Type-I, 5: small Type-II, 6: pumped storage type). |
| Comprehensive utilization | Main uses of the reservoir (mainly including power generation, water supply, shipping, flood control, and irrigation). |
| Type of regulation | Regulation types of reservoirs (mainly including day, week, season, and year). |



321        Table 2. Attributes in all (97,435) reservoirs from CRD.

| Attribute | Description |
|---|---|
| ID | Reservoir ID in this database (type: integer). |
| Name | Name of the reservoir. |
| Lat | Latitude of the reservoir point (type: float, datum: World Geodetic System (WGS) 1984, unit: °). |
| Lon | Longitude of the reservoir point (type: float, datum: World Geodetic System (WGS) 1984, unit: °). |
| Province | Province in which the reservoir is located. |
| Prefecture | Prefecture in which the reservoir is located. |
| County | County in which the reservoir is located. |
| Area | Maximum water area of the reservoir (unit: $km^2$). |
| STOR | Total storage capacity (unit: Gt). |


The Pareto distribution can describe the global distribution abundance of artificial reservoirs
and their inundation areas (sizes) (Lehner et al., 2011; Downing et al., 2006). In Figure 3, we
applied such a statistical fitting distribution to the CRD database and inferred the count of
smaller reservoirs and their total inundation area. Assuming that our data for reservoirs smaller
than 0.01 $km^2$ are complete, trend lines can be fitted and extrapolated from the Pareto
distribution to estimate smaller reservoirs not included in the CRD database. As a result, there
is an overall good fitting in the Pareto model for the CRD reservoirs in the scale of 0.01-10 $km^2$
(Figure 3a). In addition, the Pareto distributions in each basin are similar to that on the national
scale (Figure 3b-k).

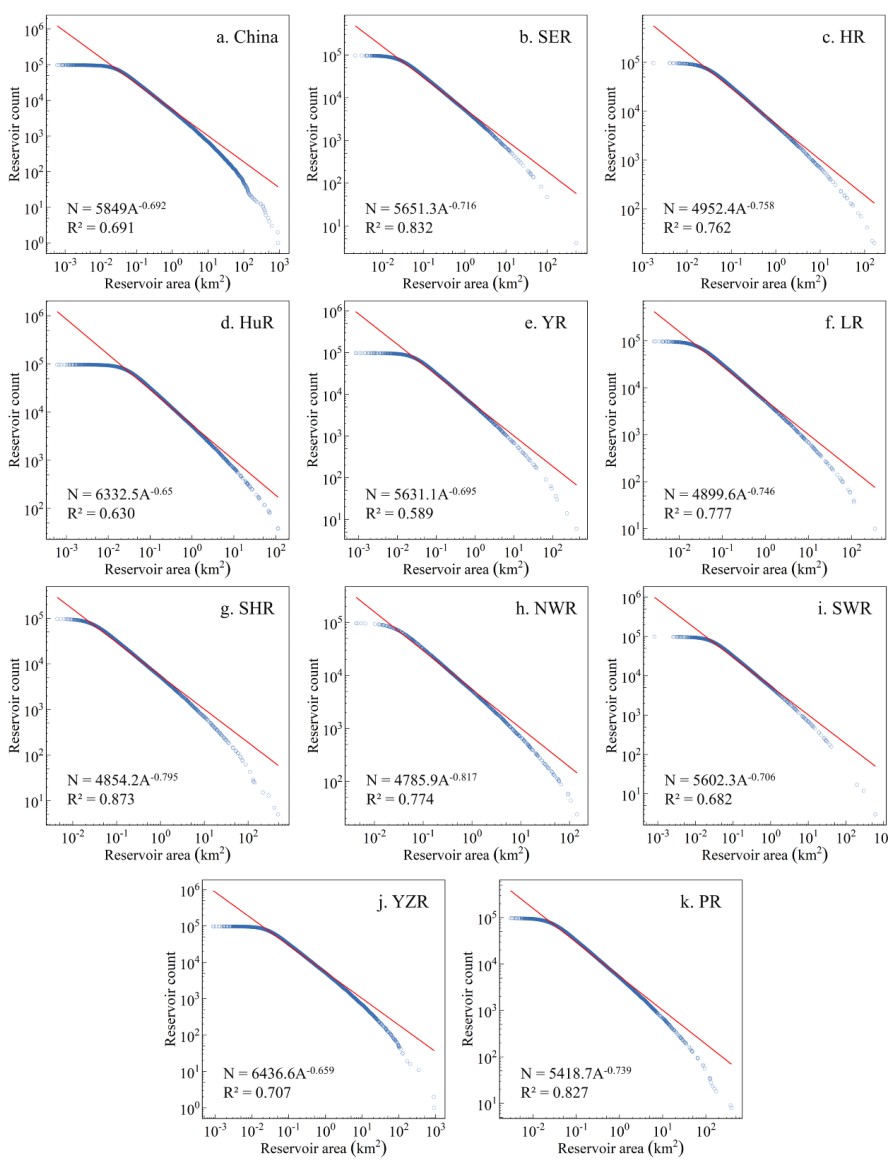

Figure 3. China reservoir area and count using a Pareto model. Distributions are plotted as the total number of reservoirs larger than a given surface area in China (a) and ten first-level water resources divisions (b-k). Blue circles intersecting the fitting lines represent the values used for model fitting.

## 4.2 Accuracy evaluation of the CRD database

To evaluate the commission and omission accuracy of the CRD database, we randomly selected


subbasin areas in each first-level river basin across China and manually checked 1,882
reservoirs. Most of them are third-level river basins. However, for the Yangtze River (YZR)
and the Yellow River (YR) basins with more reservoirs, three sub-watersheds were selected to
evenly distribute the sampled reservoirs. For each sampled reservoir, we manually confirmed
its relevant information with the recorded in the Tiandi Map. We overlapped 1,882 selected
samples with Tiandi Map for validating the geo-matching accuracy of the CRD. Then, we
manually checked whether the spatial coordinates of each sample are consistent with those
recorded in the Tiandi Map. In addition, we conducted a second round quality control to check
if any reservoirs were missing.
As shown in Table 3, the overall evaluation accuracy for the CRD database is 96.55%, ranging
from 95.47% to 98.15% in different basins. The main cause of errors in most basins is the
misclassification of "false" reservoirs (commission error), such as ponds and paddy fields. In
comparison, the accuracy was lowest in the Yangtze River basin due to the omission error of
reservoirs located in complex topographic or landscape conditions.
Table 3. Accuracy validation in each river basin.

| Region | Sample | Commission error | Omission error | Total error | Accuracy (%) |
|--------|--------|------------------|----------------|-------------|--------------|
| SER | 125 | 0 | 3 | 3 | 97.60 |
| HR | 81 | 0 | 3 | 3 | 96.30 |
| HuR | 151 | 0 | 4 | 4 | 97.35 |
| YR | 161 | 5 | 1 | 6 | 96.27 |
| LR | 162 | 2 | 1 | 3 | 98.15 |
| SHR | 69 | 2 | 0 | 2 | 97.10 |
| NWR | 177 | 5 | 0 | 5 | 97.18 |
| SWR | 45 | 1 | 0 | 1 | 97.78 |
| YZR | 685 | 4 | 27 | 31 | 95.47 |
| PR | 226 | 3 | 4 | 7 | 96.90 |

Note: SER-Southeastern River, HR-Haihe River, HuR-Huaihe River, YR-Yellow River, LR-Liaohe River,
SHR-Songhua River, NWR-Northwest River, SWR-Southwest River, YZR-Yangtze River, PR-Pearl
River. "Commission error" represents geocoding errors where the CRD information is inconsistent with
the validation reference. "Omission error" indicates the number of missing reservoirs in the samples.
**4.3 Spatial distribution of reservoirs in China**
The total area of reservoirs in China is 50,085.21 km$^2$, and the total storage capacity is estimated
to be 979.62 Gt. The spatially divergent pattern is generally characterized by the topographic
division of the Hengduan Mountains in the east-west direction and the Qinling Mountains and
the HuR in the north-south direction. The overall distribution of the reservoirs is bounded by
the Heihe-Tengchong Line that is widely recognized as a separated line for the contrasting
pattern of population, industrial development and landscape characteristics, decreasing from
southeast to northwest. Latitudinally, reservoirs in China are dominantly distributed in the belt
between 20-30°N, both in terms of count and area, whereas longitudinally, reservoirs in China
are concentrated between 100-120°E.
Chinese reservoirs are widely distributed and have obvious agglomeration characteristics.
Reservoirs are distributed not only from the hot and humid southern areas to the arid desert
areas but also from the eastern coastal areas to the Qinghai-Tibet Plateau. From Figure 4,
reservoirs are mainly distributed in China's major "Commodity Grain Production Bases" that
have a relatively great demand for agricultural irrigation, such as the Poyang Lake and Dongting
Lake Plain, HuR basin, Songnen Plain, and Sanjiang Plain. Moreover, many large reservoirs
are accumulated in areas with large elevation drops and abundant water resources. For example,
reservoirs in Sichuan province are clustered along the main stems of Fujiang River, Jialing
River, and YZR. In addition, as a major water supply, many reservoirs are concentrated in urban
areas such as the Shandong Peninsula urban agglomerations. In the Shandong Peninsula,
reservoirs are mainly concentrated in Yimeng Mountain and the Bohai Rim area.

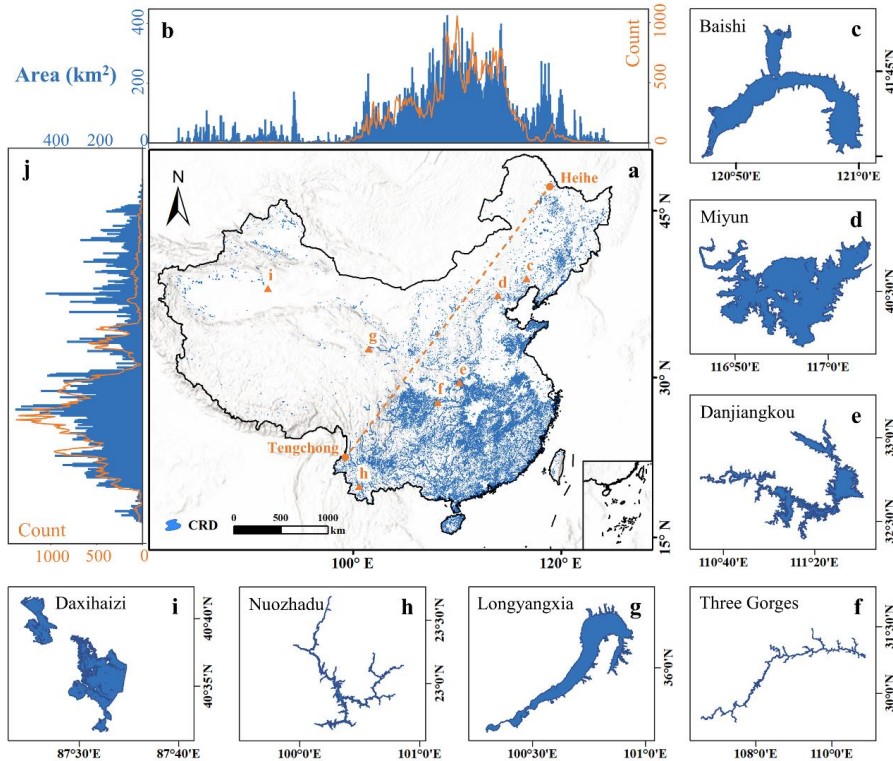

Figure 4. Spatial distribution of reservoirs in China (a). The histogram and lines represent the
area and count of reservoirs in China by 0.1° latitude (b) and longitude (j), respectively. The c-
i subgraphs show the details of Baishi Reservoir, Miyun Reservoir, Danjiangkou Reservoir,
Three Gorges Reservoir, Longyangxia Reservoir, Nuozhadu Reservoir, and Daxihaizi Reservoir.

**4.4 Distribution characteristics of reservoir storage capacity in China**

In terms of storage capacity spatial distribution, reservoirs with substantial storage capacity are
mostly found in the YZR and the PR. Many major reservoirs have been built in the SWR in
recent years, primarily in the upper stages of the Lancang, Yuan, and Nujiang rivers. The HuR
and HR basins, on the other hand, have several reservoirs, although their storage capabilities
are limited, owing to the flat terrain's minimal elevation changes. While the YR has no evident
benefit in terms of count or storage capacity, it has the biggest reservoir regulation of any basin,
and its total reservoir capacity has reached three times its annual runoff.
The distribution of reservoir storage capacity in China is shown in Figure 5. There are 135
reservoirs with a storage capacity of above 1 Gt (see Figure 5b), accounting for 60.81% of the
total. Among them, there are 15 reservoirs with a storage capacity of more than 10 Gt in China,
accounting for 29.39% of the total reservoir capacity. Also, the top 10 reservoirs (Three Gorges
Reservoir, Danjiangkou Reservoir, Longtan Reservoir, Longyangxia Reservoir, Nouzhadu
Reservoir, Xin'anjiang Reservoir, Xiaowan Reservoir, Shuifeng Reservoir, Xinfengjiang
Reservoir, and Xiluodu Reservoir) are mainly distributed in the YZR, PR, and SWR, which are
rich in water resources. These ten reservoirs alone account for 23.51% of the total storage
capacity of the CRD.

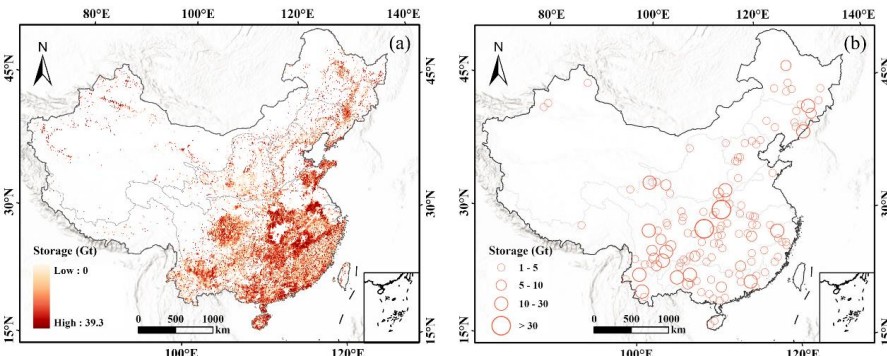

Figure 5. Distribution of reservoir storage capacity in China. Panel a shows all 97,435 reservoirs
in CRD, which are displayed in gradient color according to the total storage capacity of
reservoirs in the 0.1°×0.1° gridded statistics. Panel b shows the 135 reservoir larger than 1Gt.
Furthermore, we analyzed the distribution characteristics of reservoir number, area, and storage
capacity in each primary and secondary watershed of the water resources division. The big



bubbles illustrated in Figure 6 represent basins with a large count, large area, and large storage
capacity, which belong to the YZR. Almost all the second-level river basins with relatively large
storage capacity are distributed in the middle and upper reaches of the YZR, including the
Dongting Lake Basin, Poyang Lake Basin, the Jinsha River Basin, and the Han River Basin.

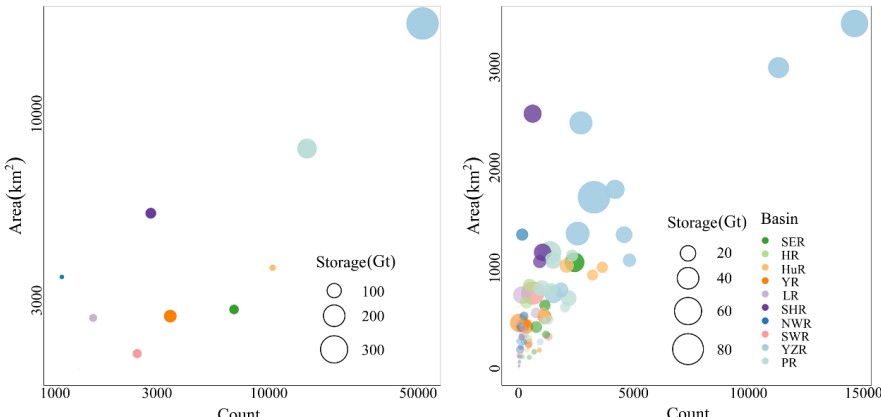


Figure 6. Bubble chart of reservoir count, area, and storage capacity of each basin in the first-
level (a) and second-level (b) hydrologic basins. Different colors represent the ten first-level
basin units. Bubble size represents the size of reservoir capacity.
**5 Discussions**
**5.1 Comparisons with other reservoir databases**
To verify the reliability of the CRD database, we compared the CRD reservoirs with the widely
recognized and publicly available reservoir/dam databases, including GOODD, GeoDAR v1.1,
and GRanD v1.3. Figure 7 shows the contrasts among these four databases. Since GOODD
does not provide reservoir attribute information (except locations and catchment areas), it is
only compared with CRD in reservoir count. The quantity of reservoirs in CRD (97,435)
exceeds those of the Chinese subsets of the global databases (from 9,238 in GOODD to 921 in
GRanD) by one to two orders of magnitude. CRD increased the total reservoir area by about
169% and 194% compared with GeoDAR and GRanD, respectively. In comparison, the total
storage capacity of CRD exceeds the GeoDAR and GRanD by 249.23 Gt and 293.51 Gt in
China, respectively. Notably, although GeoDAR still largely exceeds GRanD in dam count,
their total storage capacity was comparable, with GeoDAR increasing its reservoir storage
capacity by approximately 6% (44 Gt). This is because GRanD has included the largest
reservoirs in China.
We also compared CRD with the three global databases at different levels of reservoir areas. As
shown in Figure 8a, the advantage of CRD is most evident in the improvement of reservoirs
with an area less than 1 km$^2$, particularly reservoirs with an area less than 0.1 km$^2$. Therefore,
the total reservoir areas of the corresponding CRD database with an area smaller than 0.1 km$^2$
and 0.1-1 km$^2$ are also higher than those of other databases. For larger reservoirs (1-10 km$^2$, 10-
100 km$^2$, and larger than 100 km$^2$), the counts of CRD, GeoDAR, and GRanD have little
difference, but the CRD area is slightly higher, mainly because the reservoir polygons applied
in this study represent the maximum water extents. In addition, we found that the storage
capacity of CRD reservoirs increased at different area levels, with an average increase of 54.28
Gt. In general, CRD databases have greatly improved in terms of reservoir count, area, and
storage capacity compared with other databases in China.

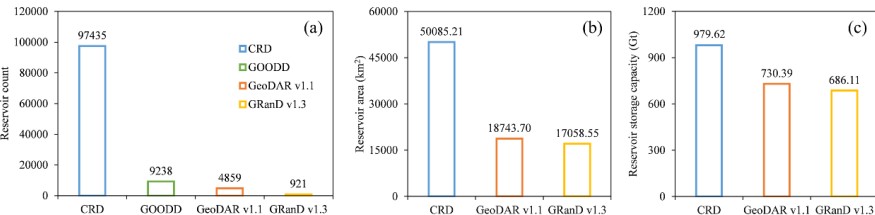


Figure 7. Comparison of reservoirs in count (a), area (b), and storage capacity (c) between the
CRD database, GOODD, GeoDAR v1.1, and GRanD v1.3 database.

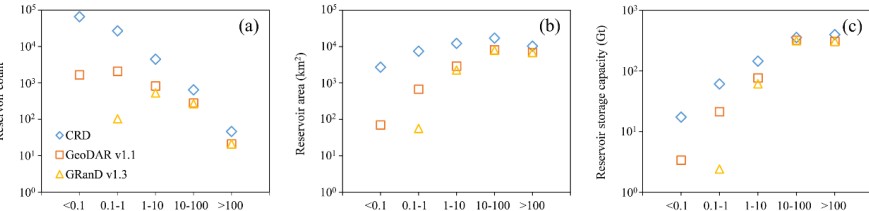


Figure 8. Comparison of reservoirs in count (a), area (b), and storage capacity (c) between the
CRD database, GeoDAR v1.1, and GRanD v1.3 database with different area levels.
**5.2 Analysis on the accumulation hotspots of the CRD reservoir distribution**
The construction of hydropower stations alleviates the energy shortage in China, reduces the
consumption of non-renewable coal energy, and makes a great contribution to the sustainable
development of China's economy and society. To further understand the characteristics of
reservoir accumulation distribution in China, we quantified the degree of reservoir
accumulation from the dimensions of the count, area, and storage, respectively.
Figure 9 shows the reservoir accumulation degree in the count, area, and storage capacity of the
CRD reservoirs. High reservoir density hotspots can be observed in the YZR's middle and lower
reaches, mainly in the Poyang Lake and Dongting Lake basins. These two-lake basins have



rugged terrains, which provide topographic convenience for constructing reservoirs. Besides,
the basins are densely populated and is an important commodity grain base, so reservoirs are
critical to meeting the agricultural irrigation water demand. The large labor force also facilitated
the reservoir construction. The construction of small and medium-sized reservoirs in China
reached a peak era under the impact of the new and old "three pillars" policy from the founding
of the People's Republic of China in 1949 to the reform and openness in 1978.
Figure 9b shows that the hotspots in the reservoir area are mainly distributed in YZR, Northeast
China, and HuR, where the terrain is relatively flat. Combined with the boom of building small
reservoirs throughout the country during the "Great Leap Forward" period, the practice of "one
piece of land for one piece of sky" even appeared in the Huaibei Plain, resulting in many
reservoirs and a large total area in the HuR. In comparison with the storage accumulation
hotspots shown in Figure 8c, we found that large reservoirs are mostly localized in the upper
reaches of the YZR and the PR. It is mainly because the Chinese reservoir construction entered
the era of big hydropower project in the 21st century. With the construction of Xiaolangdi
Reservoir, Three Gorges Reservoir, and other large hydropower stations as examples, China has
built a series of large reservoirs in the southwest of China, where there are large elevation drops
and abundant stream powers, such as the Jinsha River (the upper reaches of the YZR), the upper
reaches of the PR, and the upper reaches of the Lancang River.

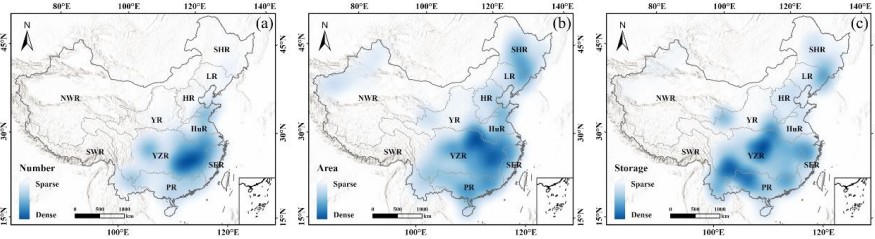

Figure 9. Distribution map of the accumulation degree of the reservoir count (a), area (b), and
storage capacity (c) in CRD.
**6 Data availability**
The China Reservoir Database (CRD) is publicly available for download from the Zenodo
repository https://zenodo.org/record/6569049 (Song et al., 2022). The database is supplied as
both shapefile format and the comma-separated values (csv) format.
**7 Conclusions**
In this study, the location information of a total of 97,435 reservoirs in China has been identified
and collected in the China Reservoir Dataset (CRD) by compiling multiple existing



dam/reservoir products, national basic geographic datasets, multi-source open map data, and
multi-level government yearbooks and database. Then, by merging three remote sensing
waterbody products, the maximum water inundation area was extracted for each of the
identified reservoirs. Based on a collection of 5,143 reservoirs with official storage capacity
records, an empirical model fitting the reservoir area-storage relationship was established to
estimate the storage capacities of other unrecorded reservoirs in CRD. The compiled reservoirs
in CRD have a total maximum inundation area of 50,085.21 $km^2$ and a total storage capacity of
about 979.62 Gt (924.96-1060.59 Gt).
Based on the CRD database, the spatial distribution characteristics of reservoir count, area, and
storage capacity were comprehensively analyzed and compared. In addition, we discussed the
improvement of CRD over other commonly-used global dam/reservoir databases and the
potential causes of several hotspots of the reservoir concentration on the context of China's
socioeconomic development and major policy implementations. The results show that
reservoirs are widely distributed across China, yet there are strong spatial heterogeneities with
several concentration hotspots. The YZR basin has the most dominant distribution in terms of
reservoir count, area, and storage capacity. Specifically, the reservoirs are mainly concentrated
in the basins of Dongting Lake, Poyang Lake, and the Han River, the middle and lower reaches
of the HuR and the YZR, the Shandong Peninsula, the Sichuan Basin, and the Yunnan-Guizhou
Plateau. The CRD database has greatly improved the reservoir mapping in terms of count, area,
and storage capacity compared with existing dam/reservoir products over the territorial area of
China. The prominent advantage of CRD could be a complete map of reservoirs smaller than 1
$km^2$. The CRD database can be used for a wide range of reservoir impact assessments and is
expected to benefit water resources management, river system investigation, hydrological
modeling, and other aspects in scientific research and sector practices.
**8 Author contribution**
CS: Conceptualization, Data curation, Formal analysis, Funding acquisition, Investigation,
Methodology, Programming, Project administration, Quality assurance, Quality control,
Supervision, Validation, Visualization, Writing – original draft preparation, and Writing –
review and editing. CF: Data curation, Formal Analysis, Investigation, Methodology,
Programming, Validation, Visualization, Quality control, Writing – original draft preparation,
and Writing – review and revision. JZ: Conceptualization, Data curation, Formal Analysis,
Investigation, Methodology, Programming, Quality control, and Writing – review and revision.
JW: Methodology, Quality control, Supervision, Validation, Writing – review and revision. YS:
Quality control, Supervision, and Writing – review and revision. KL: Quality control,



Validation, and Writing – review and revision. TC: Quality control, Validation, and Writing –
review and revision. PZ: Quality control and Validation. SL: Quality control and Validation.
LK: Quality control, Validation, and Writing – review and editing.

## 9 Competing interests

The authors declare no conflict of interest.

## 10 Acknowledgements

The authors express their gratitude to the support from the GRanD, GOODD, GeoDAR, Future
FHReD, and the national 1:250,000 public basic geographic datasets. The authors would like
to acknowledge Tiandi Map for providing a base map and the geocoding API
(https://map.tianditu.gov.cn/). The authors are also grateful to the GSW, GLAD, and FROM-
GLC10 data for providing reservoir water inundation extent, and the Almanac of China's Water
Power and other Chinese government documents for providing the reservoir storage capacity
records. This work was partly funded by the National Key Research and Development Program
of China (Grant No. 2019YFA0607101, 2018YFD0900804, 2018YFD1100101), the Strategic
Priority Research Program of the Chinese Academy of Sciences (Grant No. XDA23100102),
and the National Natural Science Foundation of China (No. 41971403).



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
