# Peer review of "A comprehensive geospatial database of nearly 100,000 reservoirs in China"

_Earth System Science Data, 2022_

## Author Comment (AC1)

Dear Editors and Reviewers:

Sincere thanks for the evaluation of this work and your valuable comments and suggestions for improving this manuscript. We carefully considered the concerning points and made efforts to improve the rigor, logic, and clarity of our manuscript titled "**A comprehensive geospatial database of nearly 100,000 reservoirs in China**". Here we submit the revised version, which has been modified according to the comments from the editor and reviewers. According to the editor and reviewers' comments/suggestions, we clarified the manuscript and response letter below regarding the appropriate paragraphs and sections. The major changes that we made in the revised manuscript are summarized as follows:

(1) To further illustrate the accuracy of the CRD database, we added a validation experiment and followed the same sampling scheme (Create Random sampling Points method) to randomly selected ten sub-basins from the remaining sub-basins, including 1,752 reservoirs. The results were added to the 'Accuracy evaluation of the CRD database' section.

(2) We added one paragraph in the 'Comparisons with other reservoir databases' section to state the contributions of the CRD database. Also, Figure 10 is added to show comparisons between GRanD v1.3, GeoDAR v1.2, GOODD, and CRD in selected regions of China.

(3) We provided the residence time information of reservoirs in the revised manuscript and database and supplemented the 'Methodology' section.

(4) As suggested, we changed the unit of reservoir storage to 'km$^3$', and updated all full names of basins.

(5) We also updated the database simultaneously. Three attributes of river order, discharge, and residence time of reservoirs were added to the revised database. The revised China Reservoir Dataset (CRD v1.1) is publicly available at https://doi.org/10.5281/zenodo.6984619.

We attach the detailed item-by-item response to all comments and suggestions for the evaluation.

Yours sincerely,

Chunqiao Song and co-authors

**COMMENTS FROM EDITORS AND REVIEWERS:**
* * *
**Referee #1:**

The manuscript entitled "A comprehensive geospatial database of nearly 100,000 reservoirs in China" proposes an improvement to the dam and reservoirs dataset by compiling existing global data and local inventories. The authors access the quality of their data and compare their data with the existing datasets, and they announce that their dataset shows great improvement and, in many respects, is better than others. The topic of the manuscript is interesting and relevant to the earth science data community, as dam dataset are critical part of earth system science. Overall, this is a well-written manuscript.

**Response:** We highly appreciate Referee #1's concise summary and positive review on the manuscript. Also, many thanks for all the constructive comments. We made changes on the manuscript with a sincere consideration of these points, the revisions and related explanations can be referred to the revised manuscript and the response letter item by item.

I have a few comments about the accuracy assessment:

1. I have concerns on the accuracy evaluation of their data (section 4.2). Line 338: does the random selection process is manual? In this section, I can not understand what accuracy means here. Do you mean area? Please provide more detail.

**Response:** Thanks for pointing out the unclear information. We corrected the statements by specifying that the random selection process is based on the Create Random Sampling Points tool from the ArcGIS Pro Data Management menu. Accuracy of the CRD database in Section 4.2 refers to the evaluation of the commission and omission errors of the database itself. Here, the commission error represents geocoding errors where the CRD information is inconsistent with the validation reference, and the omission error indicates the number of missing reservoirs in the samples.

To evaluate the commission and omission accuracy of the CRD database, we randomly selected sub-basins in each first-level river basin across China and manually checked 1,882 reservoirs. We followed the Create Random Sampling Points tool from the ArcGIS Pro Data Management menu to randomly select some subbasin areas from the first-level river basin in China. Most of them are third-level river basins. However, for the Yangtze River and the Yellow River basins with more reservoirs, three level 6 sub-basins from HydroBASINS were selected to distribute the sampled

reservoirs evenly. A total of 1,882 reservoir samples were selected, distributed in 14 sub-basins. For each reservoir sample, we manually checked whether the spatial coordinates were consistent with those recorded in the Tiandi Map. In addition, we conducted a second round of quality control to check if any reservoirs were missing. Validation results show that the overall evaluation accuracy for the CRD database is 96.55%, ranging from 95.47% to 98.15% in different basins.

To further illustrate the accuracy of the CRD database, we followed the same sampling scheme and randomly selected ten validation sub-basins from the remaining sub-basins, including 1,752 reservoirs. The distribution of all sampled validation reservoirs is shown in Figure 4. Consistent with the first validation result, the evaluation accuracy of all river basins is higher than 90%. The accuracy ranges for the CRD database from 90.70% to 97.64% among different basins, with an overall accuracy of 93.61%. Integrating the two validation results, our overall evaluation accuracy is 95.13% in terms of commission and omission errors (Table 3).

Additionally, we clarified the method for selecting sub-basins, updated Table 3, and added Figure 4 in the revised manuscript to address the referee's concern. (**Line 366-369**)

"*To evaluate the commission and omission accuracy of the CRD database, we randomly selected sub-basin areas in each first-level river basin across China and manually checked 3,634 reservoirs (Figure 4). The collection of the validation sub-basins followed the Create Random Sampling Points method.*"

[Figure]

Figure 4. The distribution of all sampled validation reservoirs.

Table 3. Accuracy validation in each river basin.

| Region | Sample | Commission error | Omission error | Total error | Accuracy (%) |
|--------|--------|------------------|----------------|-------------|--------------|
| SER | 393 | 9 | 16 | 25 | 93.64 |
| HR | 167 | 8 | 3 | 11 | 93.41 |
| HuR | 311 | 8 | 6 | 14 | 95.50 |
| YR | 289 | 12 | 4 | 16 | 94.46 |
| LR | 212 | 5 | 2 | 7 | 96.70 |
| SHR | 195 | 13 | 0 | 13 | 93.33 |
| NWR | 214 | 8 | 0 | 8 | 96.26 |
| SWR | 222 | 16 | 0 | 16 | 92.79 |
| YZR | 1278 | 28 | 29 | 57 | 95.54 |
| PR | 353 | 5 | 5 | 10 | 97.17 |

2. For dataset comparison, you can not say improvements just based on more count, area, and storage. Does this data have better accuracy over other ones? Please try to show that. Overall the accuracy assessment is not clear, which directs an unclear description of their contribution.

**Response:** We agree with the reviewer. The main purpose of CRD database is to catalogue more complete spatial distribution of reservoirs in China, especially to supplement median and small-sized reservoirs. Therefore, we cannot say that it has improvements over those global reservoir products. To clarify this point, we changed the relevant statement. Following this suggestion, we also added Figure 10 and one paragraph in the 'Comparisons with other reservoir databases' section to state the supplements of the CRD database. (**Line 481-501**)

Figure 10a shows the distribution of large reservoirs (storage capacity larger than 3 million $m^3$) in the upper reaches of the Yangtze River in GRanD v1.3, GeoDAR v1.2, and CRD. Because the GOODD dataset is limited by the basic property (reservoir storage capacity, dam height), it was not included in comparing large reservoirs. GeoDAR v1.2 incorporates GRanD v1.3 so that the pattern of large reservoirs in the upper Yangtze River is consistent between the two databases. Compared with GRanD v1.3 and GeoDAR v1.2, CRD has added 16 large reservoirs in the upper reaches of the Yangtze River, with a total storage capacity of 52.60 $km^3$, of which the total storage capacity of new reservoirs in the past five years accounted for 77.00% (40.50 $km^3$). The large reservoirs dominate the total storage capacity in the basin. Therefore, the increase of new large reservoirs dammed in recent years is one of the major differences of CRD in storage capacity.

Another supplement of CRD is to amplify the local details of smaller reservoirs based on enlarging the total area and storage capacity. Figure 10b shows the reservoir distribution in the 10-level subbasin of Poyang Lake, and Figure 10c-d contains the enlarged details in Figure 10b. GRanD v1.3, GeoDAR v1.2, GOODD, and CRD can all digitize reservoirs on rivers with catchments of more than 10 km$^2$ (Figure 10b-c). However, many smaller reservoirs were not digitized by GRanD v1.3, GeoDAR v1.2, and GOODD. Overall, CRD is relatively better mapping reservoirs at smaller watershed levels (Figure 10d).

CRD contributed to supplementing and updating new reservoirs and smaller reservoirs, nevertheless, it has a few limitations. The CRD contains a few basic reservoir attributes, such as location information (longitude, latitude, province, state, county), inundated area, and estimated water storage, and it still needs to be further supplemented and improved. Although we added reservoir residence time in the updated version, limited by the accuracy of discharge data, we only calculated the residence time of about 17,000 reservoirs.

"*Figure 10a shows the distribution of large reservoirs (storage capacity larger than 3 million m$^3$) in the upper reaches of the Yangtze River in GRanD v1.3, GeoDAR v1.2, and CRD. Because the GOODD dataset is limited by the basic property (reservoir storage capacity, dam height), it was not included in this comparison. GeoDAR v1.2 incorporates GRanD v1.3 so that the pattern of large reservoirs in the upper Yangtze River is generally comparable between the two databases. Compared with GRanD v1.3 and GeoDAR v1.2, CRD has added 16 large reservoirs in the upper reaches of the Yangtze River, with a total storage capacity of 52.60 km$^3$, of which the total storage capacity of new reservoirs constructed in the past five years accounted for 77.00% (40.50 km$^3$). The large reservoirs dominate the total storage capacity in the basin. Therefore, the increase of new large reservoirs dammed in recent years is one of the major differences of CRD in storage capacity. As shown in Figure 10b-c, GRanD v1.3, GeoDAR v1.2, GOODD, and CRD can all digitize reservoirs on rivers with catchments of more than 10 km$^2$. However, many smaller reservoirs were not compiled in GRanD v1.3, GeoDAR v1.2, and GOODD.*"

[Figure]

Figure 10. Comparisons between GRanD v1.3, GeoDAR v1.2, GOODD, and CRD in selected regions of China. Distribution of the large reservoirs (storage capacity larger than 3 million m$^3$) in the upper reaches of Yangtze River (a). Distribution of reservoirs in GRanD v1.3, GeoDAR v1.2, GOODD, and CRD in a 10-level sub-basin of Poyang Lake (b-d). Bright green triangles, orange

squares, dark green diamonds, and red dots represent GRanD v1.3, GeoDAR v1.2, GOODD, and CRD, respectively. Background image source: ESRI imagery base map.

3. What is the normal elevation in table 1. Do you mean height? It is a little confusing.

**Response:** Thanks for pointing out the confusing statement. The 'normal elevation' in the original version of Table 1 should be termed as "water level of normal storage capacity" of the reservoir. "Normal storage capacity" means that the reservoir reaches the storage capacity that can actually be used to regulate runoff.

4. Do you have residence time? Without this information, it is really hard for use in hydrological modeling?

**Response:** We appreciate the reviewer bringing this work to our attention. To follow the suggestion, we calculated the residence time of reservoirs by using river discharge data from HydroRIVERS. Moreover, we updated Table 2 and added the residence time information of reservoirs in the revised manuscript and database. (**Line 311-331**)

HydroSHEDS provides hydrographic baseline information in a consistent and comprehensive format to support regional and global watershed analyses and hydrological modeling. It is currently considered the leading global product in terms of quality and resolution (Lehner and Grill, 2013). HydroBASINS and HydroRIVERS are extracted from HydroSHEDS at a 15 arc-second resolution. HydroRIVERS represents a vectorized line network of all global rivers that have a catchment area of at least 10 km² or an average river flow of at least 0.10 m³/s, or both. HydroRIVERS contains the attribute information of each river about an estimate of long-term average discharge. Therefore, we extracted the reservoir discharge at the location of each reservoir pour point based on HydroRIVERS product. Here, the average residence time for each reservoir was calculated as the ratio between reservoir storage capacity and discharge.

The HydroRIVER dataset covers all rivers in the Pfafstetter Level 12 sub-basins of HydroBASINS, so we focused on reservoirs (17,185) that locate on these rivers, covering 96% of CRD reservoirs larger than 1 km$^2$. For the remaining reservoirs, on the one hand, they are not on the HydroRIVER rivers, and on the other hand, it is difficult to obtain the discharge of smaller reservoirs. Therefore, they are generally not included in hydrological simulations. Also, we calculated the $R^2$ of the estimated reservoir residence times and the corresponding results provided by HydroLAKES

reservoirs is 0.82. While CRD database provided information about reservoir discharge and residence time, in fact, these data can be updated as needed for specific hydrological modeling.

*"HydroSHEDS (Hydrological data and maps based on SHuttle Elevation Derivatives at multiple Scales) provides hydrographic baseline information in a consistent and comprehensive format to support regional and global watershed analyses and hydrological modeling. It is currently considered the leading global product in terms of quality and resolution (Lehner and Grill, 2013). HydroBASINS and HydroRIVERS are extracted from HydroSHEDS at a 15 arc-second resolution. HydroRIVERS represents a vectorized line network of all global rivers with a catchment area of at least 10 km² or an average river flow of at least 0.10 m³/s, or both. HydroRIVERS covers all rivers in the Pfafstetter Level 12 sub-basins of HydroBASINS and contains the attribute information of each river about an estimate of long-term average discharge. Here, we focused on reservoirs (17,185) located on HydroRIVERS rivers and extracted reservoir discharges based on HydroRIVERS. Moreover, these reservoirs cover 96% of CRD reservoirs larger than 1 km². The remaining smaller reservoirs, on the one hand, are not on the HydroRIVERS rivers, on the other hand, it is difficult to obtain the discharge of smaller reservoirs. Therefore, they are generally not included in hydrological simulations. Notably, while the CRD database provided information about reservoir discharge and residence time, these data can be updated for specific hydrological modeling. The equation of average residence time is as follows:*

$$RES\_T = \frac{V}{DIS\_AV\_CMS} \qquad (3)$$

*where DIS_AV_CMS represents the reservoir discharge in the unit of $m^3/s$, and RES_T represents the reservoir residence time in the unit of year. The $R^2$ of the estimated reservoir residence times and the corresponding results of HydroLAKES reservoirs is 0.82."*

Table 2. Attributes in all (97,435) reservoirs from CRD.

| Attribute | Description |
| --- | --- |
| ID | Reservoir ID in this database (type: integer). |
| Name | Name of the reservoir. |
| Lat | Latitude of the reservoir point (type: float, datum: World Geodetic System (WGS) 1984, unit: °). |
| Lon | Longitude of the reservoir point (type: float, datum: World Geodetic System (WGS) 1984, unit: °). |
| Province | Province in which the reservoir is located. |
| Prefecture | Prefecture in which the reservoir is located. |
| County | County in which the reservoir is located. |
| Area | Maximum water area of the reservoir (unit: $km^2$). |
| STOR | Total storage capacity (unit: $km^3$). |
| RIV_ORD | Indicator of river order using river flow to distinguish logarithmic size classes. 'RIV_ORD' refers to 'RIV_ORD' of the HydroRIVERS. |
| DIS_AV_CMS | Average long-term discharge estimate for reservoir (unit: $m^3/s$). |
| RES_T | Residence time of each reservoir (the ratio between reservoir storage capacity and discharge, unit: year). |

Note: Missing or inapplicable values are flagged by "-999".

---

## Author Comment (AC2)

Dear Editors and Reviewers:

Sincere thanks for the evaluation of this work and your valuable comments and suggestions for improving this manuscript. We carefully considered the concerning points and made efforts to improve the rigor, logic, and clarity of our manuscript titled "**A comprehensive geospatial database of nearly 100,000 reservoirs in China**". Here we submit the revised version, which has been modified according to the comments from the editor and reviewers. According to the editor and reviewers' comments/suggestions, we clarified the manuscript and response letter below regarding the appropriate paragraphs and sections. The major changes that we made in the revised manuscript are summarized as follows:

(1) To further illustrate the accuracy of the CRD database, we added a validation experiment and followed the same sampling scheme (Create Random sampling Points method) to randomly selected ten sub-basins from the remaining sub-basins, including 1,752 reservoirs. The results were added to the 'Accuracy evaluation of the CRD database' section.

(2) We added one paragraph in the 'Comparisons with other reservoir databases' section to state the contributions of the CRD database. Also, Figure 10 is added to show comparisons between GRanD v1.3, GeoDAR v1.2, GOODD, and CRD in selected regions of China.

(3) We provided the residence time information of reservoirs in the revised manuscript and database and supplemented the 'Methodology' section.

(4) As suggested, we changed the unit of reservoir storage to 'km$^3$', and updated all full names of basins.

(5) We also updated the database simultaneously. Three attributes of river order, discharge, and residence time of reservoirs were added to the revised database. The revised China Reservoir Dataset (CRD v1.1) is publicly available at https://doi.org/10.5281/zenodo.6984619.

We attach the detailed item-by-item response to all comments and suggestions for the evaluation.

Yours sincerely,

Chunqiao Song and co-authors

**COMMENTS FROM EDITORS AND REVIEWERS:**
* * *
**Referee #2:**

This manuscript describes the new dataset of most reservoirs in China (China Reservoir Dataset), including the development methodology and the characteristics of the dataset. As the reservoir is important for understanding the water resources and water risk, the constructed database is very useful for many hydrology and climate studies. The manuscript is well designed with clear explanations. I think it can be accepted after some small revisions.

**Response:** Thanks for the concise summary of this work and the highlights. The concerning points raised by the reviewer are very helpful for us to improve the manuscript. We carefully addressed these points listed below and made changes accordingly.

1. L29: 979.62 Gt

I (personally) think "$km^3$" is more common as the unit for reservoir storage.

**Response:** Thanks for the suggestion. The unit for reservoir storage is changed to '$\mathbf{km^3}$'. The usage is checked and corrected through the revised manuscript.

2. L249: water inundation extent

How is the boundary of the reservoir and connecting rivers decided from remote-sensing water extent map? Please explain

**Response:** This is a good question. Most reservoirs are formed by crossing the valley with barrages, intercepting natural river runoff, and raising the water level. Therefore, in determining the boundary between the reservoir and the connecting river, we first roughly identified the width of the upstream channel relative to that before the reservoir was built based on the high-definition images. Then, we used the topographic data to determine where the channel was widened by the water level uplift caused by the dam construction. Finally, for the last section of reservoir filling, we cut off the river that is tapered relative to the width of the river in the reservoir area by manual visual interpretation as the boundary range of the reservoir.

3. L295: SMAPE

What "SMAPE" stands for? Please spell out.

**Response:** We are sorry for missing the full term of the SMAPE (Symmetric Mean Absolute Percentage Error). The full term of the SMAPE is added in the revised manuscript. (**Line 298-300**)

"*We calculated the SMAPE (Symmetric Mean Absolute Percentage Error) of estimated storage capacity was biased of 32.62-32.64% at the 95% confidence interval based on the fitted model.*"

4. P303. Figure 3.

Why can we observe some step-wise increase in storage capacity? Please explain. (I guess the effective digits of the storage capacity data, which extent is more continuous).

**Response:** If we understand correctly, the question should be related to Figure 2. Figure 2 represents the fitting relationship between small and medium-sized reservoirs' area and storage capacity. The upper and right subplots of Figure 2 correspond to the count of reservoir area and storage capacity values, respectively. According to the concern, we carefully checked the reservoir storage capacity data used for fitting. There are two main reasons for the observed step-wise increase in reservoir storage capacity. On the one hand, we did our best to collect 4,323 recorded small and medium-sized reservoirs to establish the statistical relationship between inundation area and storage to estimate and supplement the capacity estimation of the remaining unrecorded reservoirs. While these recorded data were unevenly distributed across different reservoir levels. For example, there are 903 between 0.0001-0.0002 $km^3$ (5.00-5.30 $\log_{10}[m^3]$), accounting for 20.89%. In addition, the scale of variables will be compressed after the logarithm of the original data of storage capacity and the area is taken, making the data more aggregated. On the other hand, as the reviewer guessed, the effective digits of reservoir storage capacity data resulted in the equality of the original continuous data (see Table R1), which then led to the superposition and aggregation of sample points in Figure 2.

Table R1. The storage capacity values for the four selected reservoirs and their corresponding values after three significant decimal points.

| storage capacity values (original) | storage capacity values (three significant digits after the decimal point) |
|---|---|
| 6.081347308 | 6.081 |
| 6.080987047 | 6.081 |
| 6.080662556 | 6.081 |
| 6.080626487 | 6.081 |

5. L326: smaller than 0.01km$^2$ are complete.

This should be "larger than 0.01km$^2$".

**Response:** Thanks for noting the unclear point. We revised the statement to clarify that if our data for reservoirs larger than 0.01 km$^2$ are complete, trend lines can be fitted and extrapolated from the Pareto distribution (Figure 3 in the manuscript) to estimate smaller reservoirs not included in the CRD database.

6. L349. The main causes of errors

For user's viewpoint, the size of the lakes errors are found is better to be provided. For example, if we know there is almost no error for lakes >10 km$^2$, users can safely use the dataset for large-scale studies.

**Response**: Thanks for the suggestion. To address it, we counted the area-size levels of error reservoirs in each basin. The results indicate that 67.86% of the commission and omission error reservoirs are less than 0.10 km$^2$, and the remaining are between 0.10-1 km$^2$. This is because smaller reservoirs are more likely to be missed during manual visual inspection. However, although these validation statistics can be considered a measure of accuracy for our data products, the identified errors in the validation samples have been corrected as far as possible in our new release.

In addition, we have added relevant descriptions in the updated manuscript. (**Line 382**)

"*Also, these ponds and paddy fields are generally less than 0.10 km$^2$.*"

7. L376: YZR (and other abbreviation names)

I think you don't have to use abbreviations except for Figures and Tables in the main text. Using full name improves the readability.

**Response:** Thanks for this suggestion. All similar abbreviations in the main text are changed to full names. For example, the 'YZR' is revised to '**Yangtze River**', the 'SER' is revised to '**Southeastern River**', the 'HR' is revised to '**Haihe River**', the 'HuR' is revised to '**Huaihe River**', the 'YR' is revised to '**Yellow River**', the 'LR' is revised to '**Liaohe River**', the 'SHR' is revised to '**Songhua River**', the 'NWR' is revised to '**Northwest River**', the 'SWR' is revised to '**Southwest River**', the 'PR' is revised to '**Pearl River**'.

8. L413: Figure 6.

Please add explanations that the description of the abbreviations is found in Table 3.

**Response:** Thanks for noting the unclear points. According to the suggestion, we added the description of the abbreviations (SER, HR, HuR, YR, LR, SHR, NWR, SWR, YZR, PR) in the Figures 3 and 6 captions.

"*Note: SER-Southeastern River, HR-Haihe River, HuR-Huaihe River, YR-Yellow River, LR-Liaohe River, SHR-Songhua River, NWR-Northwest River, SWR-Southwest River, YZR-Yangtze River, PR-Pearl River.*"

**Bibliography for response letter:**

Lehner, B., Grill, G., 2013. Global river hydrography and network routing: baseline data and new approaches to study the world's large river systems. Hydrological Processes 27, 2171-2186.